# Effect of influenza vaccination on resting metabolic rate and c-reactive protein concentrations in healthy young adults

**Claire Hagan Parker**[1]*, **Srishti Sadhir**[1], **Zane Swanson**[2], **Amanda McGrosky**[1], **Elena Hinz**[1], **Samuel S. Urlacher**[3,4], **Herman Pontzer**[1,5]

**1** Evolutionary Anthropology, Duke University, Durham, North Carolina, United States of America, **2** Global Food Security Program, Center for Strategic and International Study, Washington, D.C., United States of America, **3** Department of Anthropology, Baylor University, Waco, Texas, United States of America, **4** Child and Brain Development Program, CIFAR, Toronto, Canada, **5** Duke Global Health Institute, Duke University, Durham, North Carolina, United States of America

* chparker2000@gmail.com

## Abstract

### Objectives

Chronic immune activation and severe inflammatory states are positively associated with resting metabolic rate (RMR; kcal/day), but the impacts of mild immune stimuli on metabolism are poorly understood. This study investigates the within-individual association between the inflammatory response to influenza vaccination and RMR in young adults.

### Methods

We evaluated RMRs through indirect calorimetry and circulating c-reactive protein (CRP) concentrations (mg/L)—a direct measure of inflammation—via high-sensitivity immunoassays of dried blood spots (n = 17) at baseline and two- and seven-days post-vaccine. Wilcoxon matched-pairs signed-rank tests were used to evaluate the magnitude of the CRP and RMR responses. Type II Wald chi-square tests of linear mixed-effect models assessed whether those responses were correlated.

### Results

Baseline CRP was 1.39 ± 1.26 mg/L. On day two post-vaccine, CRP increased by 1.47 ± 1.37 mg/L (p < 0.0001), representing a 106% increase above baseline values. CRP remained higher on day seven post-vaccine, 1.32 ± 2.47 mg/L (p = 0.05) above baseline values. There were no statistically significant changes in RMR from baseline to day two (p = 0.98) or day seven (p = 0.21). Change in CRP from baseline did not predict RMR variation across days (p = 0.46).

### Conclusions

We find no evidence that adult influenza vaccination results in a corresponding increase in RMR. These results suggest that the energetic cost of an influenza vaccine's mild

**Data Availability Statement:** All relevant data are within the paper and its Supporting Information files.

**Funding:** CHP URS Independent Study Grant Undergraduate Research Support Office, Duke University https://undergraduateresearch.duke.edu/opportunity/urs/urs-independent-study-grants The funders had no role in study design, data collection and analysis, decision to publish, or preparation of the manuscript.

**Competing interests:** The authors have declared that no competing interests exist.

inflammatory stimulus is either too small to detect or is largely compensated by a temporary downregulation of energy allocated to other metabolic tasks.

## Introduction

Both maintaining a functioning immune system and mounting an immune response are metabolically expensive. For example, activation of the immune system can elicit "sickness behavior," an adaptive response characterized by undernutrition and sedentary activity [1,2]. By downregulating the digestive system function and physical activity and switching to adipose tissue as fuel, the body frees up valuable energy resources to devote to immune responses [2–4]. However, when immune activation is sufficiently high, the body no longer compensates for increases in the immune system's energy expenditure by decreasing energy allocation to other systems and increases whole body metabolic rate. Dramatic increases in resting metabolic rate (RMR, kcal/day) have been observed in people experiencing severe inflammatory states or chronic immune system activation [1,5–9].

The magnitude of immune system activation necessary to elicit a change in RMR is still unknown. One study found that the metabolic cost of a mild immune stimulus in mice was compensated for by reductions in energy allocation to the reproductive and digestive systems [10]. Another observed an 8% increase in RMR and a 10% decrease in testosterone while male university students had mild respiratory infections [11]. To date, only two studies have examined changes in RMR due to vaccination. Both evaluated responses to typhoid fever vaccination for eight hours immediately following injection and observed elevations in RMR that were mediated by increases in body temperature [12,13]. Because these studies stopped measuring RMR at the eight-hour mark, it is unclear if RMR continued to rise, how long it remained elevated, or if it correlates with circulating concentrations of acute-phase response proteins like c-reactive protein (CRP).

CRP serves as a reliable surrogate measure of innate immune system activity because it stimulates the production of proinflammatory cytokines and is a direct measure of inflammation [14,15]. Its concentration in the blood can rise by 100,000% above baseline during severe inflammatory states [16]. Concentrations return to baseline within a couple of days because CRP has a relatively short half-life of about nineteen hours and is quickly cleared by the kidneys [15,17]. Additionally, several cross-sectional and longitudinal studies have found that elevated CRP is correlated with decreased rates of growth in children; these findings suggest that there is a tradeoff between the energetic cost of CRP production, or closely associated immune processes, and the energetic demands of growth [7–10,18].

Several studies have utilized influenza vaccines as a model of mild stimulation of the immune system and CRP as a sensitive measure of inflammation induced by that stimulus [14,19–22]. Such studies have consistently found that after vaccination, CRP concentrations peak in two to three days and approach baseline five to seven days post-vaccination [20–22]. Mounting such a response requires energy investment [2,5,11–13]. However, the magnitude of the investment and its impact on metabolism is unknown. By using a within-individual design to examine the relationship between RMR and CRP in response to mild immune system stimulation by influenza vaccination, this project aims to gain further insight into how humans evolved to allocate energy resources during immune events.

## Methods

The study was approved by the Duke University Health System Institutional Review Board. In September and October of 2021, we recruited participants from Durham, North Carolina

communities through an online interest form. Respondents received an eligibility and avail-ability survey via email. To be eligible, prospective participants had to be willing to receive the influenza vaccine, be fully vaccinated for COVID-19, and be at least 18 years old. Potential par-ticipants who had already received a 2021–22 influenza vaccine, were pregnant, took any med-ications that could affect metabolic rate, or had a chronic illness were excluded. Ten females and seven males living in Durham, North Carolina gave written informed consent and partici-pated in the study. One participant did not complete the final, day seven post-vaccine mea-surement due to acute illness. Four participants reported mild cold-like symptoms on at least one visit.

Participants were scheduled to come for data collection at the same time of day at visit. For the RMR measurements to be accurate, they were instructed to fast and avoid exercise for at least eight hours before their visit. At the beginning of each visit, participants reported the time of their last caloric intake, and their fasting status was confirmed via a blood glucose mea-surement. Baseline blood samples, vitals, and RMRs were obtained in the morning before par-ticipants got their influenza vaccines. After leaving the lab, participants received their vaccines and sent corresponding documentation to study personnel. Further blood samples, vitals, and RMR measurements were collected two and seven days after vaccination. All data was de-iden-tified during collection.

The same anthropometric and physiological measurements were taken at each visit, exclud-ing blood lipid profiles which were only measured once. Subject height was measured on a SECA stadiometer. Weight, body water percentage, and body fat percentage were measured using a Tanita BF-680W biometric digital scale. Blood pressure and heart rate were measured on an Omicron 7 automated BP cuff before and after each RMR measurement. Body tempera-ture was measured on the Braun ThermoScan Ear Thermometer.

RMR was measured using the COSMED Quark RMR metabolic cart. The metabolic cart measures oxygen consumption ($VO_2$), carbon dioxide production ($VCO_2$), and the fraction of expired carbon dioxide ($FeCO_2$), among other ventilatory parameters. RMR (kcal/day) is esti-mated using the modified Weir equation RMR (kcal/day) = [(3.941 × $VO_2$ + 1.106 × $VCO_2$) × 1.44] [23]. The device was calibrated with calibration gas and for flow rate at the beginning of every day measurements were taken. Measurements took place in a dimly lit, thermoneutral, and quiet room.

Participants lay supine for twenty-five minutes on an examination bed with a transparent ventilated canopy hood over their heads, necks, and upper chests. A plastic sheet attached to the canopy hood covered the participant and the examination bed to separate the air inside the hood from the air in the room. Both the hood and the plastic sheet were sanitized after each measurement. A digital turbine flowmeter measured ventilation, and gas was sampled at the expiratory port of the ventilated hood at 10-second intervals. The oxygen and carbon dioxide concentrations of the samples were analyzed to determine the metabolic rate (kcal/day). We adjusted the pump speed on the cart throughout the measurement to keep the $FeCO_2$ between 0.70 and 1.10%. Participants were instructed to be as still as possible, breathe normally, and relax while remaining awake for the duration of the measurement. The first five minutes of every twenty-five-minute measurement were automatically discarded to allow participants to reach a steady state. For each participant, the five-minute period with the lowest average RMR from the remaining twenty minutes was used for analysis [24]. Among the RMR measure-ments, two participants' baseline measurements and another's day two measurement were invalid based on Cook's distance [25] and excluded from analyses.

Blood samples were collected as finger prick capillary whole blood spots under standard antiseptic procedures on Whatman 903[TM] Protein Saver Cards. The cards were dried at room temperature for at least four hours before being placed into plastic bags with desiccant packs

and stored at -20°C until shipped to the Human Evolutionary Biology and Health Lab at Baylor University for analysis and continued -20°C storage. CRP concentration was determined using a modified high-sensitivity ELISA protocol developed specifically for dried blood spot samples [26]. All samples were measured in duplicate, with resulting intra- and inter-assay coefficient variations of 1.84% and 1.44%, respectively, and a lower limit of detection of 0.03mg/L. After the blood spots were collected, participants' blood glucose concentrations and blood lipid profiles were measured via finger prick using CardioChek ST Analyzers with PTS test strips.

All analyses were conducted with the R statistical programming language [27]. The study sample's characteristics and clinical parameters were summarized using means ± standard deviation. We used Wilcoxon matched-pairs signed-rank tests to perform non-parametric tests of differences in CRP and RMR between visits. Type II Wald chi-square tests of linear mixed effect models were implemented to evaluate whether blood glucose and CRP could be used to predict RMR. We log-transformed (base 10) CRP results to normalize the distribution.

## Results

Demographic and anthropometric data are reported in Table 1, and all vitals are reported in Table 2. Participants had an average temperature of 36.3 ± 0.4°C; none were febrile at any point. Participants fasted for an average of eleven hours prior to each visit and had fasting blood glucose concentrations of 98 ± 13 mg/dL.

Participants' RMRs and CRP measurements are reported in Table 3. The mean baseline CRP for the entire sample was 1.39 ± 1.26 mg/L, while median was 1.09 (0.50–1.76) mg/L. On day two after vaccination, one participant had a CRP concentration greater than the 13.435 mg/L upper detection limit. This CRP measurement was excluded from analysis. As shown in Fig 1, the highest CRP concentrations were observed two days after vaccination with a mean of 2.69 ± 2.05 mg/L and a median of 2.10 (1.33–3.62) mg/L. CRP significantly increased from baseline to day two by an average of 1.47 ± 1.37 mg/L and a median of 1.24 (0.50–1.64) mg/L (Wilcoxon matched-pairs signed-rank test; $p < 0.0001$). On day seven after vaccination, the mean CRP was 2.76 ± 2.77 mg/L, while median CRP was 2.06 (0.74–3.56) mg/L. The average elevation from baseline was 1.32 ± 2.47 mg/L, while the median elevation from baseline was 0.23 (0.00–1.47) mg/L ($p = 0.05$). Day two and day seven CRP levels were right-skewed, with a few participants exhibiting elevated CRP far above the IQR (Fig 1).

At baseline, the mean RMR was 1406 ± 309 kcal/day, while median RMR was 1409 (1287–1592) kcal/day. On days two and seven after vaccination, the mean RMRs were 1443 ± 297 kcal/day and 1488 ± 312 kcal/day, respectively, while median RMRs were 1408 (1299–1571) kcal/day and 1440 (1268–1656) kcal/day, respectively. There were no statistically significant changes in RMR from baseline to day two ($p = 1$) or day seven ($p = 0.12$). Distributions of RMR on each visit, controlling for fat-free mass (FFM), are shown in Fig 2.

**Table 1. Baseline demographics of study participants (n = 17).**

| Measures | Mean ± Standard Deviation |
|---|---|
| Age (years) | 20.9 ± 1.6 |
| Sex (% female) | 58.2 |
| Height (cm) | 169.5 ± 10.3 |
| Weight (kg) | 66.6 ± 13.2 |
| Body Fat (%) | 22.0 ± 7.5 |

**Table 2. Vitals of study participants (n = 17).**

| Vitals | Baseline | Day Two | Day Seven |
|---|---|---|---|
| Blood Glucose (mg/dL) | 98.9 ± 11.2 | 98.6 ± 15.2 | 95.8 11.6 |
| Temperature (˚C) | 36.4 ± 0.5 | 36.3 ± 0.4 | 36.3 ± 0.4 |
| Total Cholesterol (mg/dL)* | 141.0 ± 22.9 | n/a | n/a |
| HDL (mg/dL)* | 79.0 ± 15.4 | n/a | n/a |
| Triglycerides (mg/dL)* | 116.2 ± 35.3 | n/a | n/a |
| LDL (mg/dL)* | 39.2 ± 19.5 | n/a | n/a |
| Pre-RMR Systolic Blood Pressure (mmHg) | 118.0 ± 12.6 | 117.3 ± 13.5 | 116.1 ± 10.2 |
| Pre-RMR Diastolic Blood Pressure (mmHg) | 80.2 ± 7.7 | 77.5 ± 9.3 | 78.4 ± 8.0 |
| Post-RMR Systolic Blood Pressure (mmHg) | 113.0 ± 10.0 | 112.4 ± 13.8 | 113.4 ± 13.6 |
| Post-RMR Diastolic Blood Pressure (mmHg) | 79.2 ± 8.3 | 76.8 ± 8.6 | 78.6 ± 7.9 |
| Pre-RMR Heart Rate 1 (bpm) | 74.2 ± 12.4 | 74.5 ± 12.7 | 73.1 ± 12.5 |
| Post- RMR Heart Rate 2 (bpm) | 66.2 ± 8.7 | 65.1 ± 11.0 | 67.3 ± 12.0 |

Values are mean ± standard deviation. HDL, high-density lipoprotein; LDL, low-density lipoprotein.

* Blood lipid concentrations were only evaluated once and measurements that were below the detection limits of the analyzer were not included in calculations of means and standard deviations.

Type II Wald chi-square tests of the linear mixed effect model using only log(CRP) as a fixed effect showed that it did not significantly predict RMR (p = 0.47). The linear models using log(CRP) (p = 0.45) and blood glucose level (p = 0.77) as fixed effects revealed that neither **significantly** predicted RMR. Finally, the model using FFM and log(CRP) (p = 0.41) showed that only FFM (p < 0.001) was a significant predictor of RMR.

**Table 3. RMR and CRP at baseline, two days, and seven days after influenza vaccination.**

| Participant | Baseline RMR (kcal/day) | Day two RMR (kcal/day) | Day seven RMR (kcal/day) | Baseline CRP (mg/L) | Day two CRP (mg/L) | Day seven CRP (mg/L) |
|---|---|---|---|---|---|---|
| 1 | 1480 | 1487 | 1514 | 1.98 | 3.34 | 2.08 |
| 2 | 1638 | 1683 | 1766 | 0.28 | 1.48 | 0.29 |
| 3 | 1146 | 1265 | 1016 | 0.50 | 1.86 | 0.24 |
| 4 | 1078 | 1040 | 1202 | 3.70 | 8.69 | 2.63 |
| 5 | 713* | 1499 | 1323 | 1.09 | 4.01 | 2.46 |
| 6 | 1365 | 1393 | 1420 | 4.05 | >13.435** | 4.02 |
| 7 | 2001 | 2075 | 2059 | 0.15 | 0.64 | 0.38 |
| 8 | 1409 | 965* | 1521 | 0.63 | 4.53 | 0.74 |
| 9 | 1305 | 1206 | 1284 | 1.28 | 2.89 | 7.69 |
| 10 | 1487* | 1942 | 1970 | 1.76 | 3.49 | 0.95 |
| 11 | 1125 | 1215 | 1162 | 0.28 | 1.57 | 0.82 |
| 12 | 1575 | 1423 | 1461 | 0.42 | 0.85 | 6.88 |
| 13 | 1927 | 1796 | 1962 | 3.51 | 4.05 | 8.88 |
| 14 | 1268 | 1322 | 1269 | 0.50 | 1.01 | 0.73 |
| 15 | 1350 | 1310 | 1264 | 1.26 | 1.44 | 2.04 |
| 16 | 1608 | 1534 | 1619 | 1.67 | 2.34 | 3.41 |
| 17 | 1432 | 1376 | | 0.56 | 0.87 | |

* These three RMR measurements were invalid based on Cook's distance and excluded from analysis.

** The CRP concentration in this sample was greater than the 13.435 mg/L upper detection limit of the high-sensitivity immunoassay.

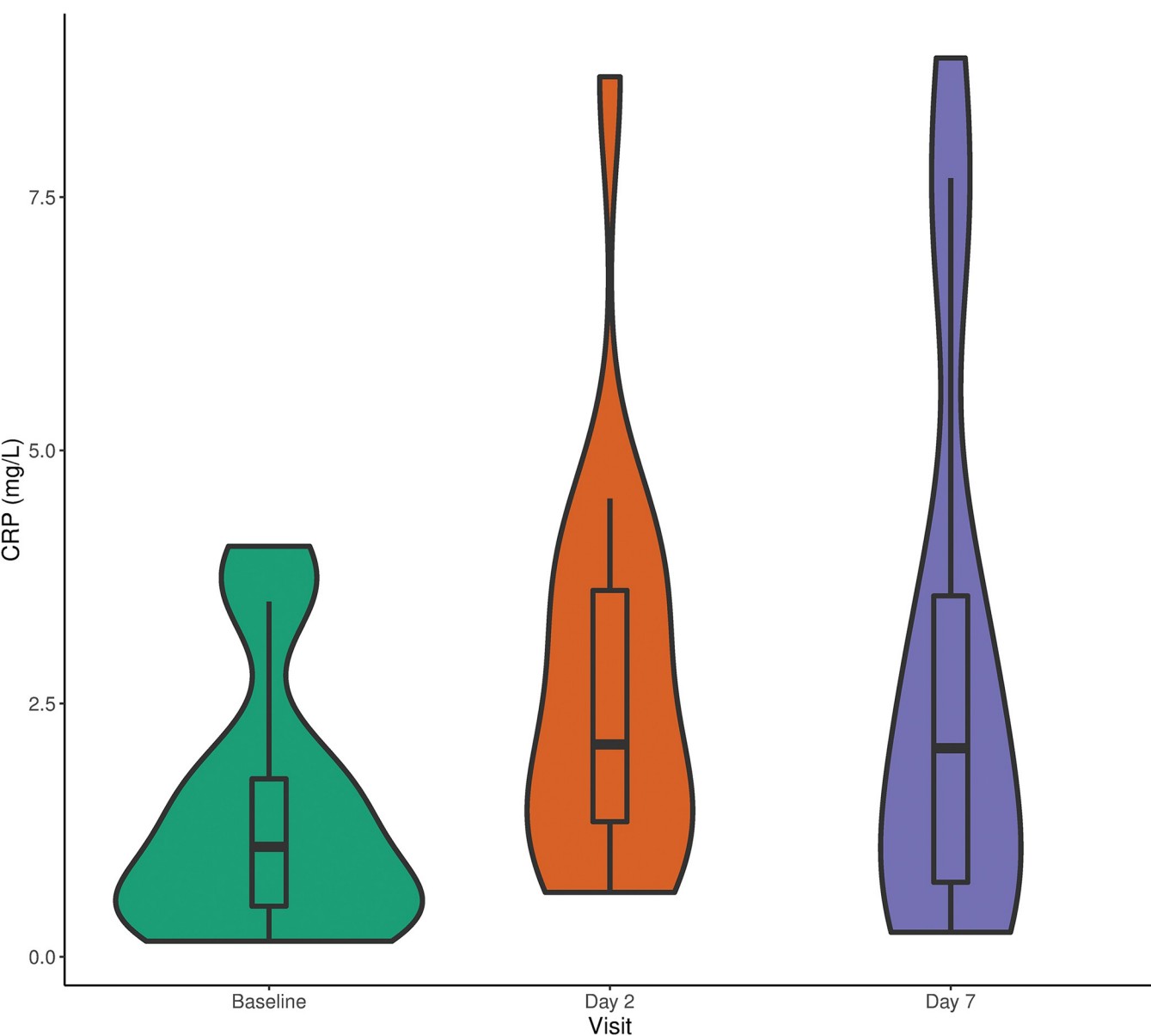

**Fig 1. Density plot of CRP at baseline, two days, and seven days after influenza vaccination.** Density distributions plots of CRP (mg/L) measured at baseline before influenza vaccination and two and seven days after influenza vaccination.

## Discussion

While the impacts of chronic immune activation and severe inflammatory states on metabolism are well documented [1,5–9,18,28], our understanding of the effect that a mild immune stimulus has on metabolic rate is limited. To the best of our knowledge, this study is the first to investigate the relationship between CRP and RMR in response to influenza vaccination. Results from this study suggest that influenza vaccination is too small of an immune stimulus to induce an elevation in RMR. While our results do not suggest that influenza vaccination impacts RMR, they do corroborate prior research findings that influenza vaccination induces a mild acute inflammatory response that can be measured by CRP.

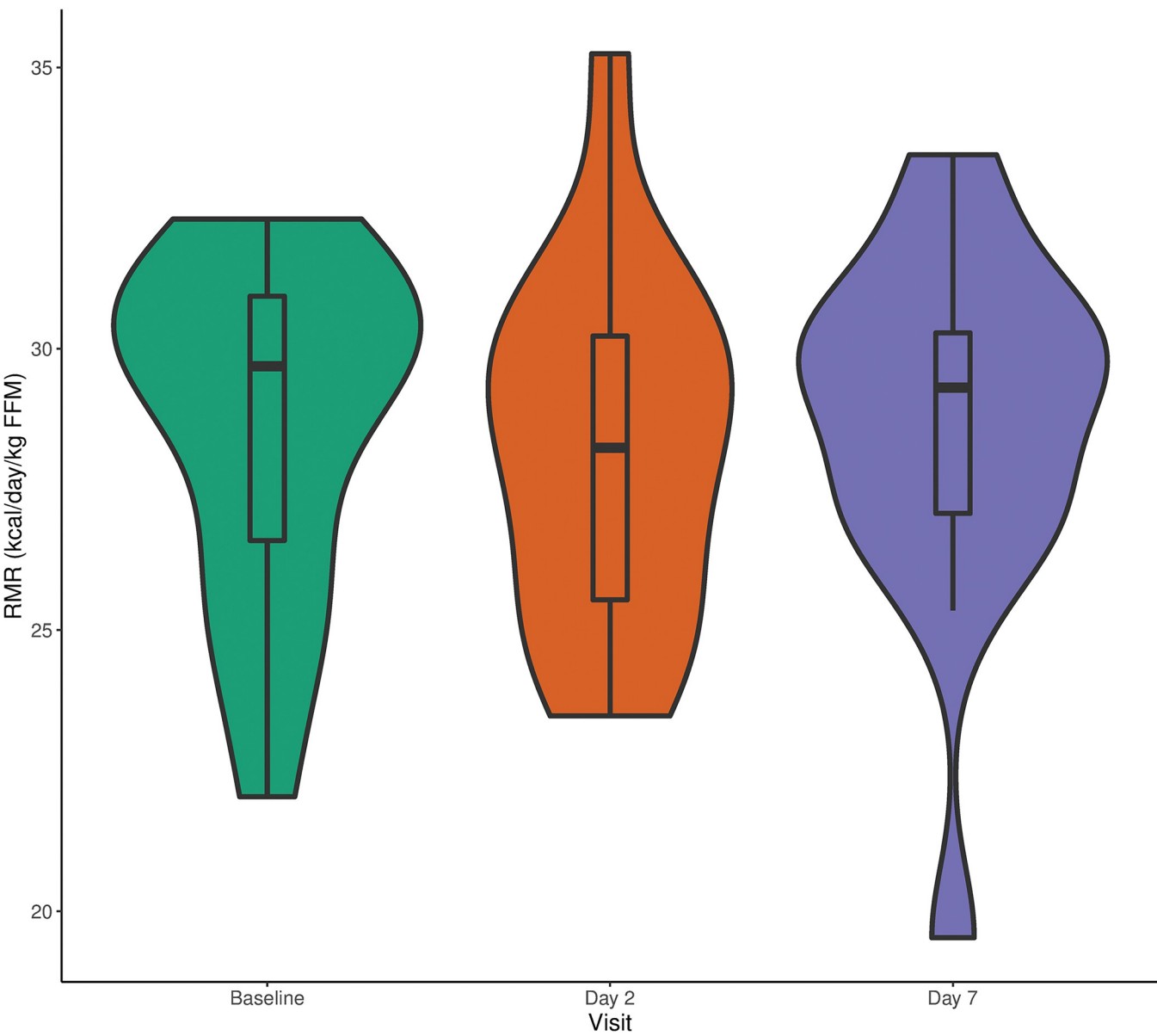

**Fig 2. Density plot of RMR at baseline, two days, and seven days after influenza vaccination.** Density distributions plots of RMR, controlling for fat-free mass, measured at baseline before influenza vaccination and two and seven days after influenza vaccination.

## CRP increases in response to influenza vaccination

Our sample of healthy young adults had a median baseline CRP of 1.09 mg/L which is in line with population-level research in the United States that has found 20-29-year-olds to have a median baseline of 1.4 mg/L [29]. Two days after vaccination, we observed a median increase in CRP of 1.24 mg/L, representing a median 112% elevation from baseline—much larger than the 30–40% elevations from baseline reported in most prior studies [14,20,21]. Compared to a Seattle-based study, our response was smaller than the mean 227% increase observed in men with carotid artery disease (CAAD) and greater than the mean 63% increase in men without CAAD [19]. Furthermore, we observed sustained elevations in CRP—a median of 54% above

baseline—seven days after vaccination. In contrast, Tsai and colleagues reported CRP **concentrations** only 7% above baseline seven days following influenza vaccination [21].

The effect of age on CRP response is the most likely explanation for our sample's larger CRP response to influenza vaccination. The average age of our sample, 20.9 years, is much lower than that of the comparable studies mentioned above that tended to focus on older adults. Age impacts both baseline CRP and the magnitude of CRP elevation in response to an immune system activation [29–31]. Lane-Cordova and colleagues reported that even compared to elderly adults with low baseline CRP concentrations, young adults experienced a greater elevation in CRP from baseline after receiving an influenza vaccine [30]. Another possible explanation is that some participants could have become infected with a pathogen during their participation in our study. This would result in our measured CRP responses to vaccination being artificially high because infection with an actual pathogen would induce a larger magnitude CRP response [17,32].

## RMR does not increase in response to vaccination

We found no evidence that influenza vaccination impacted RMR; there were no trends nor statistically significant differences in RMR across the three measurements. Moreover, a study investigating the day-to-day variability of RMR measurements using a ventilated hood system determined that an observed change in RMR must be at least 6% to represent a clinically meaningful change in metabolic rate [33]. In our study, the mean RMRs (1406 ± 309 kcal/day, 1443 ± 297 kcal/day, and 1488 ± 312 for baseline, day two, and day seven, respectively) were all under that 6% threshold, so these differences are neither statistically nor clinically significant. As such, it appears that influenza vaccination was too weak of an inflammatory stimulant to trigger an increase in RMR in our sample.

Previous research has found that large or chronic stimulations of the immune system are so energetically costly that they are not fully compensated for by downregulating energetic investment in other body systems [1,5–9,18,28]. The present study aimed to fill a gap in the existing literature by investigating how a mild immune system stimulus impacts metabolism. Our results suggest that the energetic cost of influenza vaccination may be compensated for by the body reallocating energetic resources from one or several body systems to the immune system. The body's immune response to influenza vaccination occurs over a relatively short period and appears to be minimal in magnitude. Therefore, brief, and minor, energetic divestment from other metabolic activities would presumably not have lasting consequences and upregulation of the basal metabolic rate would not occur. In mice, a mild immune stimulus was not associated with changes in metabolic rate, but it was associated with reductions in energy allocated to digestive and reproductive systems [10]. We did not evaluate reproductive or digestive function in our participants, but future studies should investigate whether changes in those systems are associated with mild immune stimuli.

One possible explanation for why influenza vaccination was too mild of an immune stimulus to result in a change in RMR is that all participants had received influenza vaccinations in the past. Participants already had memory B-cells with the blueprint to synthesize antibodies against some of the proteins in the vaccine. As a result, the adaptive immune system would have been able to respond more quickly, while the innate immune system would not require robust activation [2,34].

Also, previous research has found that RMR and CRP concentrations are correlated at baseline [35], but, given that influenza vaccination is a mild immune stimulus, changes in CRP may have been too small to result in measurable increases in RMR. In severe inflammatory states like sepsis where 30% increases in RMR have been observed, CRP concentrations can

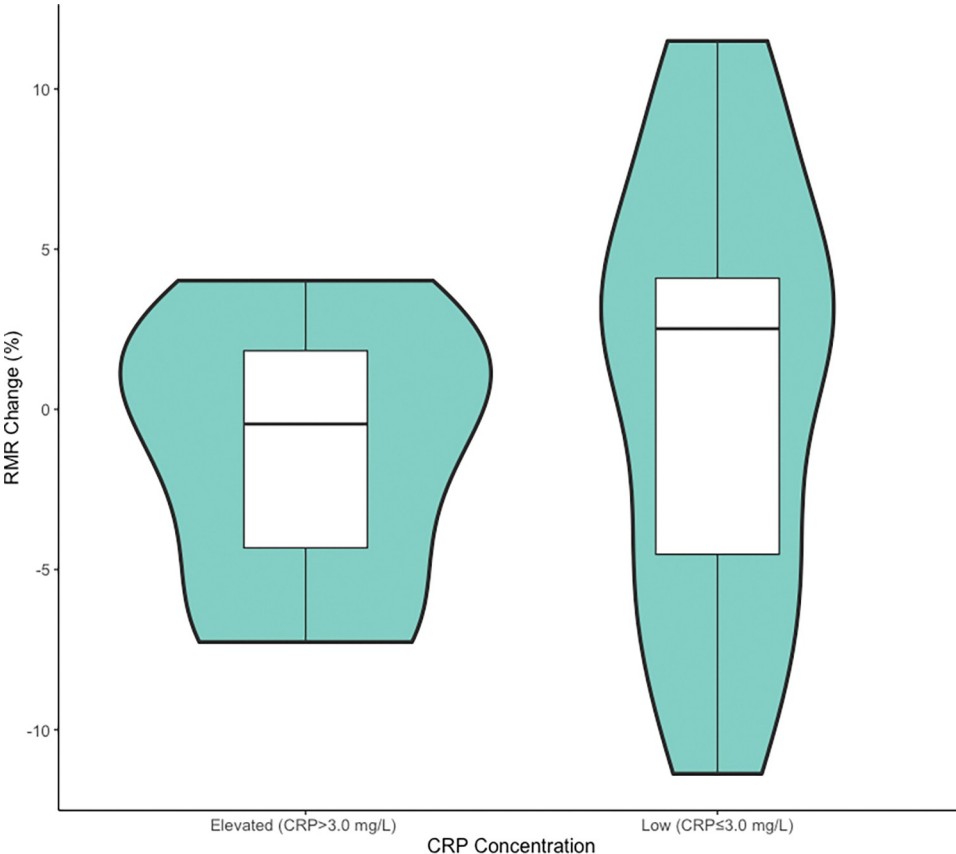

**Fig 3. Density plot of percent change in baseline RMR after influenza vaccination stratified by CRP concentration.** Stratification of density distribution plots of percent change in baseline RMR after influenza vaccination by CRP concentration. RMR was measured at baseline (before vaccination) and at days two and seven after vaccination; the day two:baseline and day seven:baseline ratios were combined for analysis. CRP measurements on days two and seven after vaccination were combined and subdivided as Elevated (CRP>3.0 mg/L) or Low (CRP≤3.0 mg/L).

increase by 100,000%, whereas we observed a median 112% increase in CRP two days after influenza vaccination [5,6,36]. Furthermore, any increases in RMR associated with the immune response to influenza vaccination would likely be too small to measure and would be obscured by statistical noise, especially given our relatively small sample size (n = 17). Individuals with CRP levels between 3.0 mg/L and 10.0 mg/L are considered to have low-grade inflammation [15,21,26]. As such, we considered participants with CRP>3.0mg/L as having elevated CRP. When we tested the association between RMR and CRP among participants with elevated CRP (>3.0 mg/L), there was still no correlation (p = 0.47), shown in Fig 3. This result suggests that CRP concentrations would have to be extremely high, like they are in severe inflammatory states, before accompanying increases in RMR could be observed.

If influenza vaccination impacts RMR but RMR peaks at a different time than CRP, the RMR response would not have been captured within our study design. We based the timing of our RMR measurements on when we expected CRP to peak (two days after influenza vaccination) using existing literature, but it is possible that changes in RMR would occur at a different time than changes in CRP. The only two studies that have investigated the effect of vaccination on RMR found that RMR increased much earlier in response to typhoid fever vaccination i.e., 7–16% at six to eight hours after injection [12,13]. These studies stopped evaluating RMR after

six to eight hours, so it is unknown whether this timeframe following injection represents the peak RMR response. Influenza vaccination may cause a similar increase in RMR several hours following injection, but we would not have observed the change with the current study design.

## Conclusions and future directions

Our results corroborate previous findings that influenza vaccination is a valid model for investigating inflammatory responses to mild immune system stimuli *in vivo*. We expand the existing literature on CRP reactivity with our finding that healthy young adults appear to experience larger acute increases in CRP in response to mild immune system stimulation than older adults, as reported in previous studies. We also add to the existing research on the energetic cost of immune system activation. In contrast to sepsis and chronic activations of the immune system, we find no evidence that mild acute immune system activation induces a corresponding increase in adult RMR. Rather, our results suggest that energetic costs associated with mild acute inflammatory responses are either too small to be measured by indirect calorimetry or are largely compensated for by temporary reallocations of energy from other body systems.

Future studies evaluating the energetic costs of mild immune system activation studies should consider using whole-room calorimetry to monitor any changes in metabolic rate as well as alterations in dietary intake, activity level, and sleep duration [37]. Further, future studies should evaluate the immune response in vaccine naïve patients (i.e., not previously exposed to a given vaccination). Other vaccinations beyond influenza (e.g. tetanus, tick encephalitis, COVID-19, etc.) may elicit different or stronger immune responses and should also be considered when designing future studies. Examining vaccine response in other populations, particularly those with greater background pathogen burden, would also advance our understanding of immune activity costs.

## Supporting information

**S1 Dataset. Study raw data.**
(XLSX)

## Acknowledgments

We thank J.P. Dunn, Ph.D. and C.L. Nunn, Ph.D. for their feedback on this project.

## Author Contributions

**Conceptualization:** Herman Pontzer.

**Data curation:** Claire Hagan Parker.

**Formal analysis:** Claire Hagan Parker.

**Funding acquisition:** Claire Hagan Parker, Herman Pontzer.

**Investigation:** Claire Hagan Parker, Srishti Sadhir, Zane Swanson, Amanda McGrosky, Elena Hinz, Samuel S. Urlacher.

**Methodology:** Claire Hagan Parker, Herman Pontzer.

**Project administration:** Claire Hagan Parker, Zane Swanson, Herman Pontzer.

**Resources:** Zane Swanson, Samuel S. Urlacher, Herman Pontzer.

**Supervision:** Srishti Sadhir, Zane Swanson, Herman Pontzer.

**Visualization:** Claire Hagan Parker.

**Writing – original draft:** Claire Hagan Parker.

**Writing – review & editing:** Srishti Sadhir, Zane Swanson, Amanda McGrosky, Elena Hinz, Samuel S. Urlacher, Herman Pontzer.

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
