## [Decision Letter · Decision Letter 0]

22 Aug 2023

PONE-D-23-11732Effect of influenza vaccination on resting metabolic rate and c-reactive protein concentrations in healthy young adultsPLOS ONE

Dear Dr. Parker,

Thank you for submitting your manuscript to PLOS ONE. After careful consideration, we feel that it has merit but does not fully meet PLOS ONE’s publication criteria as it currently stands. Therefore, we invite you to submit a revised version of the manuscript that addresses the points raised during the review process.

We look forward to receiving your revised manuscript.

Kind regards,

Juan J Loor

Academic Editor

PLOS ONE

Journal Requirements:

“We thank the Undergraduate Research Support Office at Duke University for financially supporting this project. We thank J.P. Dunn, Ph.D. and C.L. Nunn, Ph.D. for their feedback on this project.”

“CHP

URS Independent Study Grant

Undergraduate Research Support Office, Duke University https://undergraduateresearch.duke.edu/opportunity/urs/urs-independent-study-grants

Reviewers' comments:

Reviewer's Responses to Questions

**Comments to the Author**

1. Is the manuscript technically sound, and do the data support the conclusions?

Reviewer #1: Yes

2. Has the statistical analysis been performed appropriately and rigorously? 

Reviewer #1: Yes

3. Have the authors made all data underlying the findings in their manuscript fully available?

Reviewer #1: Yes

4. Is the manuscript presented in an intelligible fashion and written in standard English?

Reviewer #1: Yes

5. Review Comments to the Author

Reviewer #1: In ”Effect of influenza vaccination on resting metabolic rate and c-reactive protein concentrations in healthy young adults” the authors investigate the impact of a mild immune stimulus (influenza vaccination) on changes in resting metabolic rate (RMR) and circulating c-reactive protein (CRP) concentrations as a marker of ongoing inflammatory processes. While severe inflammation or chronic immune activation poses a high energetic demand enhancing the resting metabolic rate, there are only a few studies investigating the connection between the level of inflammation and the bodily functions concerned to cover its energetic cost. In case of a mild inflammation caused by influenza vaccination the authors did not find association between the lightly elevated CRP levels and changes in RMR; Parker et al. suggest further studies on the changes in dietary intake, activity level, and sleep duration complementing the investigation of changes in metabolic rate for finding the source of energy covering the expenditures of immune system activation.

Strength: The study was carried out in a comprehensive cohort, investigation methods are chosen wisely and using R for statistical analysis is appealing. Even following an as mild stimulation as an influenza vaccination the authors detected elevation in CRP level while the resting metabolic rate did not change.

Limitations: 1. All the participants of the study received influenza vaccination in earlier years. It would have been interesting to include participants without any such experience. Other kind of vaccination patients could have been included as well to gain a more nuanced picture, e.g. patients receiving tetanus vaccination, tick encephalitis vaccination or others. 2. As the authors also pointed out in the discussion, RMR was reported to peak at six to eight hours after vaccination but no measurements were taken at that time.

Major issues:

1. While in the text mean CRP levels are mentioned, Figure 1 shows the median CRP level instead. CRP values are not normally distributed after the stimulus.

2. Figure 3 is not informative. The authors should discuss in further details what the result on the plot shows and how does it affect the drawn consequences. Did the authors test if and how the changes in the ratio of CRP levels correlate with the changes in the ratio of RMR? Showing the differences on the axes of the plot can be misleading since these values are dependent on the scale of the measurement.

3. In Figure 4, patients are devided based on CRP level. Elevated CRP level was defined as CRP>3.0 mg/ml. Could the authors rationalize this choice? Is this CRP level at day 0 (before vaccination)? On the plot are the day 2/ day 0 and day 7/ day 0 ratios combined?

Minor issues:

1. In line 190, does BGL refer to blood glucose level? Please open the abbreviation.

2. It is misleading to read about a „1.0-fold increase” (line 265) or „an average increase in CRP of 1.47 mg/L, representing a 106% elevation from baseline” (line 212).

6. PLOS authors have the option to publish the peer review history of their article (what does this mean?). If published, this will include your full peer review and any attached files.

Reviewer #1: **Yes: **Zsuzsanna Ortutay

---

## [Author Response · Author response to Decision Letter 0]

31 Oct 2023

Response to Reviews

Reviewer 1

The study was carried out in a comprehensive cohort, investigation methods are chosen wisely and using R for statistical analysis is appealing. Even following an as mild stimulation as an influenza vaccination the authors detected elevation in CRP level while the resting metabolic rate did not change.

Author response: Thank you!

1. All the participants of the study received influenza vaccination in earlier years. It would have been interesting to include participants without any such experience. Other kind of vaccination patients could have been included as well to gain a more nuanced picture, e.g. patients receiving tetanus vaccination, tick encephalitis vaccination or others.

Author response: We agree that this is a limitation of our study. We have included an additional sentence in the Conclusions and future directions section to suggest future studies evaluate multiple types of vaccines.

The revised text reads as follows on lines 317-321: 

“Further, future studies should evaluate the immune response in vaccine naïve patients (i.e., not previously exposed to a given vaccination). Other vaccinations beyond influenza (e.g. tetanus, tick encephalitis, COVID-19, etc.) may elicit different or stronger immune responses and should also be considered when designing future studies.”

2. As the authors also pointed out in the discussion, RMR was reported to peak at six to eight hours after vaccination but no measurements were taken at that time.

Author response: We agree that this is a limitation of our study. However, we believe we have sufficiently addressed it in our discussion and conclusions and future directions sections. Our study design aligned with previous studies evaluating the effect of influenza vaccination on CRP, where peak CRP is expected two days post-vaccination. The only studies of vaccination effects on RMR have been conducted with the typhoid vaccine, which differs in several ways from the influenza vaccine. Therefore, we choose to follow the existing influenza vaccination literature when designing this study. It may be worth developing future studies to measure RMR much earlier after injection.

3. While in the text mean CRP levels are mentioned, Figure 1 shows the median CRP level instead. CRP values are not normally distributed after the stimulus.

Author response: We agree with the reviewer’s assessment. Accordingly, throughout the manuscript, we have reported both mean ± SD and median (IQR) CRP and RMR levels to account for the non-normal distribution. Additionally, we included the following text on lines 170-171:

“Day two and day seven CRP levels were right-skewed, with a few participants exhibiting elevated CRP far above the IQR (Fig. 1).”

4. Figure 3 is not informative. The authors should discuss in further details what the result on the plot shows and how does it affect the drawn consequences. Did the authors test if and how the changes in the ratio of CRP levels correlate with the changes in the ratio of RMR? Showing the differences on the axes of the plot can be misleading since these values are dependent on the scale of the measurement.

Author response: We agree with the reviewer’s assessment that Figure 3 did not add valuable information to our paper and have decided to remove it from the manuscript.

5. In Figure 4, patients are divided based on CRP level. Elevated CRP level was defined as CRP>3.0 mg/ml. Could the authors rationalize this choice? Is this CRP level at day 0 (before vaccination)? On the plot are the day 2/ day 0 and day 7/ day 0 ratios combined?

Author response: As suggested by the reviewer, we clarified our reasoning for defining the elevated CRP level as CRP>3.0 mg/ml. The revised text reads as follows on lines 276-278:

“Individuals with CRP levels between 3.0 mg/L and 10.0 mg/L are considered to have low-grade inflammation [15,22,27]. As such, we considered participants with CRP>3.0mg/L as having elevated CRP” 

We also updated the language in the description of the figure (now Figure 3) to clarify how variables were used for analysis. The revised text reads as follows on lines 287-289:

“RMR was measured at baseline (before vaccination) and at days two and seven after vaccination; the day two:baseline and day seven:baseline ratios were combined for analysis. CRP measurements on days two and seven after vaccination were combined and subdivided as Elevated (CRP>3.0 mg/L) or Low (CRP≤3.0 mg/L).”

6. In line 190, does BGL refer to blood glucose level? Please open the abbreviation.

Author response: We opened the abbreviation of BGL to blood glucose level (line 196).

7. It is misleading to read about a „1.0-fold increase” (line 265) or „an average increase in CRP of 1.47 mg/L, representing a 106% elevation from baseline” (line 212).

Author response: We replaced references to fold increases with percentage increases in the introduction (line 63) and the discussion (line 272).

---

## [Decision Letter · Decision Letter 1]

23 Nov 2023

Effect of influenza vaccination on resting metabolic rate and c-reactive protein concentrations in healthy young adults

PONE-D-23-11732R1

Dear Dr. Parker,

We’re pleased to inform you that your manuscript has been judged scientifically suitable for publication and will be formally accepted for publication once it meets all outstanding technical requirements.

Kind regards,

Juan J Loor

Academic Editor

PLOS ONE

Additional Editor Comments (optional):

Reviewers' comments:

Reviewer's Responses to Questions

**Comments to the Author**

1. If the authors have adequately addressed your comments raised in a previous round of review and you feel that this manuscript is now acceptable for publication, you may indicate that here to bypass the “Comments to the Author” section, enter your conflict of interest statement in the “Confidential to Editor” section, and submit your "Accept" recommendation.

Reviewer #1: All comments have been addressed

2. Is the manuscript technically sound, and do the data support the conclusions?

Reviewer #1: Yes

3. Has the statistical analysis been performed appropriately and rigorously? 

Reviewer #1: Yes

4. Have the authors made all data underlying the findings in their manuscript fully available?

Reviewer #1: Yes

5. Is the manuscript presented in an intelligible fashion and written in standard English?

Reviewer #1: Yes

6. Review Comments to the Author

Reviewer #1: (No Response)

7. PLOS authors have the option to publish the peer review history of their article (what does this mean?). If published, this will include your full peer review and any attached files.

Reviewer #1: **Yes: **Zsuzsanna Ortutay

---

## [Editor Report · Acceptance letter]

7 Dec 2023

PONE-D-23-11732R1 

Effect of influenza vaccination on resting metabolic rate and c-reactive protein concentrations in healthy young adults 

Dear Dr. Parker:

I'm pleased to inform you that your manuscript has been deemed suitable for publication in PLOS ONE. Congratulations! Your manuscript is now with our production department. 

Kind regards, 

on behalf of

Dr. Juan J Loor 

Academic Editor

PLOS ONE